# IMPACT: A Large-scale Integrated Multimodal Patent Analysis and Creation Dataset for Design Patents

**Homaira Huda Shomee**   **Zhu Wang**   **Sourav Medya**   **Sathya N. Ravi**
Department of Computer Science, University of Illinois Chicago
{hshome2,zwang260,medya,sathya}@uic.edu

## Abstract

In this paper, we introduce IMPACT (**I**ntegrated **M**ultimodal **P**atent **A**nalysis and **Crea​T**ion Dataset for Design Patents), a large-scale multimodal patent dataset with detailed captions for design patent figures. Our dataset includes half a million design patents comprising 3.61 million figures along with captions from patents granted by the United States Patent and Trademark Office (USPTO) over a 16-year period from 2007 to 2022. We incorporate the metadata of each patent application with elaborate captions that are coherent with multiple viewpoints of designs. Even though patents themselves contain a variety of design figures, titles, and descriptions of viewpoints, we find that they lack detailed descriptions that are necessary to perform multimodal tasks such as classification and retrieval. IMPACT closes this gap thereby providing researchers with necessary ingredients to instantiate a variety of multimodal tasks. Our dataset has a huge potential for novel design inspiration and can be used with advanced computer vision models in tandem. We perform preliminary evaluations on the dataset on the popular patent analysis tasks such as classification and retrieval. Our results indicate that integrating images with generated captions significantly improves the performance of different models on the corresponding tasks. Given that design patents offer various benefits for modeling novel tasks, we propose two standard computer vision tasks that have not been investigated in analyzing patents as future directions using IMPACT as a benchmark viz., 3D Image Construction and Visual Question Answering (VQA). To facilitate research in these directions, we make our IMPACT dataset and the code/models used in this work publicly available here.

## 1 Introduction

Design patents are important for intellectual property protection in industries where aesthetics and user experience are predominant, such as consumer electronics, fashion, and automotive design [27, 1]. Unlike utility patents, which protect the functional aspects of an invention, design patents safeguard the ornamental or aesthetic features of a product. This includes elements such as shape, surface ornamentation, and overall visual appearance. Design patents provide inventors with exclusive rights to their creations and prevent others from producing, selling, or using designs that are substantially similar. In the past few years, the number of design patents granted in the US is increasing. Though these are publicly available by law, the data is not well-organized for patent analyses which could help patent offices as well as can serve as a benchmark for the machine learning community.

**Existing works on patent datasets.** A variety of patent data formats, including XML, TSV, TIFF, and PDF, are publicly accessible for bulk download from multiple sources. One notable dataset, USPTO2M [33], comprises approximately 2 million utility patents specifically curated for classification tasks. Another dataset, BIGPATENT [48], contains a corpus of 1.3 million utility patents, exclusively in textual format. Additionally, the Harvard University Patent Dataset (HUPD) [55]

38th Conference on Neural Information Processing Systems (NeurIPS 2024) Track on Datasets and Benchmarks.

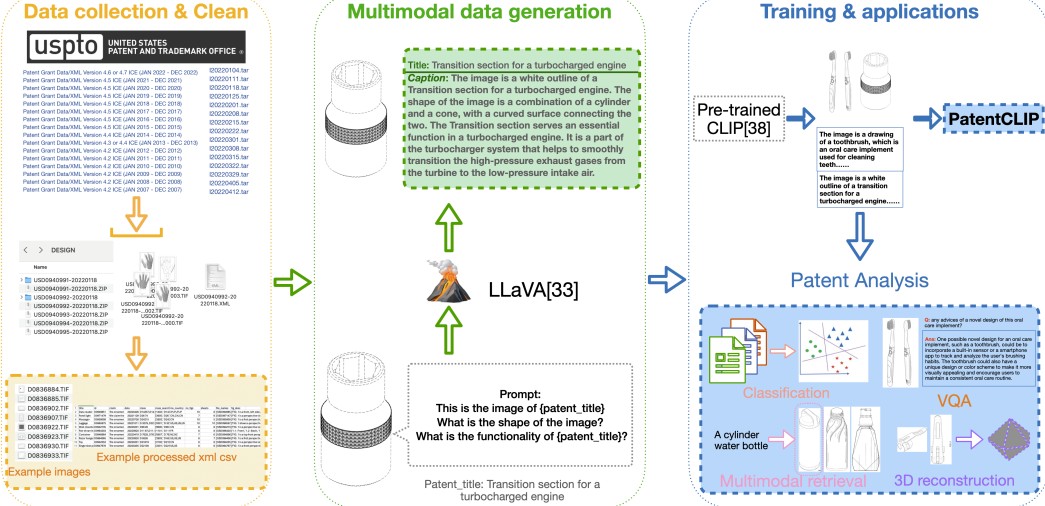

Figure 1: The main components of IMPACT dataset. We collect and pre-process the raw data to construct a comprehensive well-structure and accessible design patent dataset. We integrate multimodal information and design various training and application scenarios.

consists of a broader range of data fields, compiling 4.5 million patents intended for multipurpose patent analysis. *These datasets focus only on utility patents.* Despite the presence of figures in utility patents, these datasets omit the visual data, concentrating solely on text to support research in natural language processing (NLP). Among few works on *design patents*, DeepPatent [27] has curated a dataset focused on patent drawings, which is designed to enhance the retrieval of technical illustrations. This dataset includes 45,000 unique design patents, covering the year 2018 and the first six months of 2019. DeepPatent2 [1] has introduced a dataset with over 2.7 million technical drawings extracted from 14 years of U.S. design patent documentation spanning 2007 to 2020. This dataset includes brief uninformative captions that mention only the viewpoints of the figures. In contrast, our dataset is larger and has richer information with 11 fields and elaborated captions.

**Limitations of existing datasets.** The limitations of the existing works are as follows.

- **Single modality:** Existing datasets primarily focus on either text-based or image-based data and thus, neglects the integration of both modalities. This singular approach restricts their applicability in a comprehensive patent analysis and other multimodal tasks.
- **Lack of large & organized design patent datasets:** Most datasets consist of utility patents. The only existing two design datasets are either small or do not include data from recent years. They also lack other meta information.
- **Lack of important information:** Since design patents do not include detailed descriptions of the figures, the absence of descriptive captions makes patent analysis difficult. This is particularly impactful for tasks such as searching for prior art or work, which are essential for preventing patent infringement. Current datasets do not include such descriptive information.

**Our contributions, IMPACT:** To address these limitations, we develop a new large dataset IMPACT that integrates both textual and visual information from the design patents within a unified framework. Our major contributions are as follows.

- **Multimodality:** We introduce a multimodal patent dataset that includes patent images, metadata, and detailed captions to support a variety of NLP, vision, and multimodal tasks. This dataset is also valuable for patent analysis tasks such as classification, retrieval, prior art searches, and design trend analysis.
- **Comprehensive dataset:** We have compiled a collection of 435,101 patents spanning 16 years from U.S. design patent documents. This extensive collection includes a total of 3,609,805 drawing figures. Additionally, our dataset consists of eleven fields such as the title, patent ID, claims, date of publication, classification code, and extensive image-related information, including the number of images per patent and descriptions of the viewpoints.

- **Descriptive captions**: To address the absence of descriptions about the designs, such as features and shapes, we generate elaborated captions by employing a vision-language model. It generates descriptive captions for the design figures, capturing details from the sketch. These captions, coupled with the images, enrich our dataset and becomes a valuable resource for advanced patent analysis and multimodal research applications.

  **Code and data.** The codebase and data are available at the following link: https://github.com/AI4Patents/IMPACT.

## 2 Background and Related Work

### 2.1 Background on Design Patents

A patent is a legal document granting exclusive rights to an inventor for a specific period and it allows the inventor to exclude others from making, using, or selling their invention[1]. Specifically, we focus on design patents and it has limited information compared to utility patents. While a utility patent is composed of a title, an abstract, multiple detailed claims, a thorough description, and drawings, along with some metadata; the design patents are more limited in textual content consisting of only a title, a single claim, and small descriptions of the figures, alongside drawings of the invention. The claim in a design patent typically starts with "The ornamental design for..." and does not include the extensive, detailed claims that are rich in textual data found in utility patents. The descriptions of the figures cover different viewpoints such as perspective, top, front, rear, bottom, and enlarged details, as depicted in the drawings (please see examples in Figure 7 in the Appendix). These drawings are presented on drawing sheets, where sometimes one sheet may display multiple figures, thus showing different viewpoints in a composite figure. Beyond this, there is no additional text data to describe the images, their shapes, or functionality, since these figures lack any sort of captions.

**Challenges.** While the utility patents have already elaborated set of information, working with design patents are challenging due to several reasons. *First*, design patents protect the ornamental aspects of a product and they are inherently subjective. Determining whether a new design infringes on an existing design patent often involves interpreting visual as well as aesthetic elements and this process is more difficult than with the utility patents. *Second*, the scope of a design patent is determined by the drawings and these drawings must be meticulously accurate. However, the quality of these drawings are often not suitable for the existing machine learning models to use. *Third*, design patents only protect the appearance of an object, not its functional aspects. This is limited as these patents do not have any other information. We overcome this by incorporating elaborated captions in this dataset (see Section 3). To combat these challenges, we believe curating a unified dataset for patent analyses and other machine learning tasks (e.g., VQA) will be extremely useful.

### 2.2 Related Work on Patent Analysis

**Patent classification.** One of the major patent analysis tasks is assigning CPC or IPC codes[2] (utility patents) or design codes[3] (design patents) to the submitted patents, which is time-consuming due to the numerous classification codes and their hierarchical structure. Various models have been proposed in the literature to automate this process and they can be categorized into three major types [50]. *First, the traditional methods* share a two-step approach: generating initial features followed by using a classifier. For instance, [14] use a single-layer LSTM with Word2Vec features for IPC subgroup classification. Similarly, [46] employ LSTM with fixed hierarchy vectors for IPC subclass classification. [42] and [43] train fastText embeddings on 5 million patents and used Bi-GRU for classification. [53] extracts key patent sections, trained Word2Vec, and used parallel LSTMs. *Second, the ensemble models* use different word embeddings and deep learning techniques. [6] uses SVM as a baseline, experimenting with various datasets, features, and semi-supervised learning. [21] and [22] explore ensemble models with Bi-LSTM, Bi-GRU, LSTM, and GRU, focusing on word embeddings and partitioning techniques, respectively. In contrast, patent images are classified into visualization types using the CLIP model with MLP and various CNN models in [13]. *Third, among the LLM-based approaches*, as the first study, [30] fine-tunes the BERT model on the USPTO-2M

---

[1]https://www.wipo.int/patents/en/
[2]https://www.cooperativepatentclassification.org/cpcSchemeAndDefinitions
[3]https://learn.library.wisc.edu/patents/lesson-3/

dataset. Similarly, [44] fine-tunes BERT, XLNet, and RoBERTa on USPTO-2M, establishing XLNet as the new state-of-the-art with the highest precision, recall, and F1 measure. [2] use domain-adaptive pre-training with Linguistically Informed Masking, showing that SciBERT, pre-trained on scientific literature, outperforms BERT in patent classification.

**Patent retrieval.** The patent retrieval task [26, 23, 7, 45] aims to efficiently retrieve relevant patent documents and images based on search queries. This is important for identifying new patents, evaluating novelty, and avoiding infringement. Patent image retrieval can also inspire design. *First, traditional machine learning* methods for patent retrieval, [45] outlines five technical requirements for AI feasibility, including query expansion and identifying semantically similar documents. [54] uses random forest, Support Vector Regression, and Decision Trees to merge search results effectively. *Second, neural network-based methods* have recently gained popularity for patent retrieval. [26], [20], and [27] use CNN, DUAL-VGG, and ResNet, respectively, for retrieving patent images based on query images. *Third, among LLM-based methods,* [40] utilizes CLIP for image embedding and RoBERTa for textual features, enhancing searches with visual and textual data. [23] uses BERT with combinations of title, abstract, and claim. [51] employs SBERT for text embeddings and TransE for citation and inventor knowledge graph embeddings and finds that mean cosine similarity among patent vectors effectively links multiple existing patents to a target patent.

**Patent quality analysis.** Businesses usually evaluate patent value due to its impact on revenue and investment [4] and investors aim to predict the future value of technological innovations when making decisions. Consequently, companies hire professional analysts for quality analysis as it requires substantial effort and domain expertise [34]. MLP-based approaches utilizing several indices have been employed in the past [12, 56]. These indices usually include claim counts, forward citations, backward citations etc. [32] classifies patents based on their maintenance period into four categories using a Bi-LSTM with attention mechanism and Conditional Random Field (CRF) to assess patent quality in its initial stages. [18] predict forward citations and investor reactions to patent announcements using CNN-LSTM neural networks and various ML models. [8], [34], and [4] apply neural networks such as CNN, Bi-LSTM, Attention-based CNN (ACNN), and deep and wide Artificial Neural Networks (ANN), respectively. Additionally, large models such as a variation of BERT (MSABERT) has been also employed to assess patent value based on the textual data [25].

**Patent generation.** Recently, as generative models are gaining popularity, they have been used to reduce the need for human effort in writing long patent documents. [29] uses GPT-2 to generate independent patent claims, fine-tuning on patent claims from USPTO. This method generates the first claim given a few words but lacks quantitative metrics for evaluating claim quality. [28] focuses on personalized claim generation by fine-tuning GPT-2 with inventor-centric data, using BERT to assess the relevance of generated claims to the inventor.

Most of the above approaches are applied to *utility patents* where the patents have detailed information. One of the major reasons that the *design patents* are not well-studied is because of the lack of suitable datasets. Our work aims to address this gap by building a comprehensive dataset.

## 3    Our IMPACT Dataset

Our dataset, IMPACT includes 435,101 design patents issued by the USPTO between January 2007 and December 2022 with a total of 3,609,805 images. We provide a discussion of the data collection methodology, data format, structure, and key statistical attributes of the dataset. Additionally, we discuss the limitations associated with the dataset. Please see more details on the dataset and its analyses in the Appendix.

**Data Collection & Processing.** US patent data is publicly available, but it is stored in a complex format and not suitable for usual machine learning tasks. Thus, we collect and organize the data into an unified and accessible format for NLP and Computer Vision (CV) tasks. We have obtained the data from the USPTO Bulk Data Storage System (BDSS)[4]. The patent grant full-text data with embedded TIFF Images is published weekly as tar files, containing data for all three types of patents—utility, design, and plant patents. Each design patent includes one XML file accompanied by multiple TIFF images, with each image describing different angels of the design. More specifically, we have collected the data for design patents to process and construct our dataset.

---

[4]https://bulkdata.uspto.gov/

Each design patent is accompanied by an XML file that contains all the metadata for that particular patent. Our text processing pipeline involves parsing plain text from the XML documents to extract the essential metadata. This process includes identifying and isolating key elements such as titles, claims, and figure descriptions.

**Data Format & Fields.** From the XML files, we have collected and created 11 necessary fields and stored them in year-wise CSV files. All the images are kept in their original form, i.e., TIF format, as provided by the USPTO. For each design patent, the dataset has a separate folder named using the format USDID-date. For example, *USD0534331-20070102*, where *D0534331* is the document number/ID and *20070102* is the date. We have included 11 fields in the dataset. These are as follows: title, ID, claim, date, classification, US field of classification search[5], applicant or inventor country, number of figures for the design, number of drawing sheets, names of image files, and descriptions of figures. Detailed descriptions with examples of all fields are provided in Table 5 ( App. A.1).

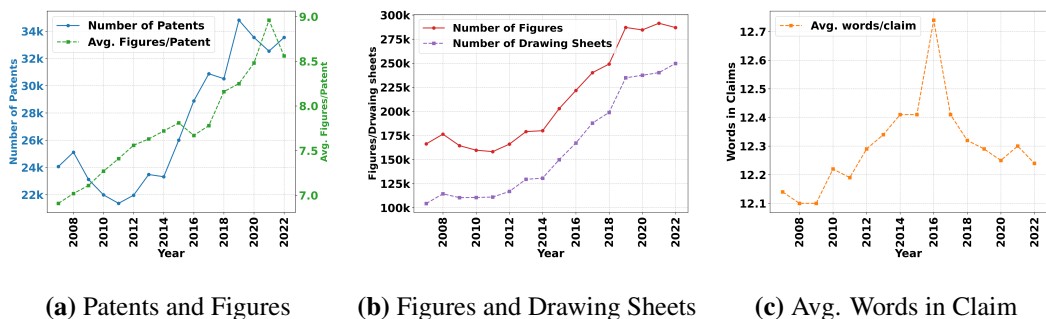

**(a)** Patents and Figures     **(b)** Figures and Drawing Sheets     **(c)** Avg. Words in Claim

Figure 2: Patent statistics (2007-2022): (a) Number of patents and average number of figures per patent, (b) Number of figures and drawing sheets. (c) Average number of words in a claim. These show a general increase in both the number of patents and figures.

**Quantitative Analyses.** Figure 2 illustrates the patent statistics in terms of the number of design patents published annually and the corresponding figures in them. Figure 2a shows line charts depicting the trend in the number of patents over time and average number of figures per patent. The data shows a general trend of increase which suggests that not only are more patents being granted each year, but the complexity or detail of these patents, as indicated by the number of figures, is also rising. The data also reveals that the average number of figures per patent ranges between 7 and 9 indicating that patents are incorporating more viewpoints and detailed illustrations to enhance the clarity and understanding of a design. Figure 2b details the growth in the total number of figures associated with all patents and the number of drawing sheets used. This increase is consistent with the growing volume of patents. Figure 2c shows the number of words in each patent claim. Design patent has only one single line claim and it does not provide detailed information about the drawing. This necessitates generating richer information (e.g., an elaborated caption) for the design. Additional quantitative analyses are provided in Appendix A.1.

**Caption generation.** Design patent images typically lack detailed captions, which can be highly beneficial for various NLP and multimodal tasks [39, 19, 47, 57]. This dataset can contribute as a benchmark for the existing multi-modal tools as well as it can also help in patent analyses (see Sec. 4). To create such a dataset, we add descriptive captions of the images in our dataset. For the caption generation task, we use LLaVA (Language-Image Assisted Visual Analysis) [35], GPT-4o[6], and Qwen-VL [5] to generate captions for 1000 samples from the data of 2022. We consider three factors for choosing caption models including runtime, cost, and quality of the captions. In terms of runtime, GPT4-o, LLaVA, and Qwen-VL required an average of 4.7, 3.98, and 3.25 seconds to generate each caption, respectively. For costs, GPT-4o is not free, whereas LLaVA and Qwen-VL are free of charge. Regarding caption quality, qualitative examples (See Fig. 8 in Appendix A.2) reveal that Qwen-VL often fails to provide functional descriptions and sometimes includes Chinese words in English captions. We also conduct zero-shot multimodal retrieval tasks to evaluate the quality of captions (See Table 1). GPT-4o performs best and LLaVA outperforms Qwen-VL. Considering all

---

[5]https://www.uspto.gov/patents/laws/examination-policy/seven-classification-design-patents
[6]https://openai.com/index/hello-gpt-4o/

the above factors including cost, we use LLaVA as our captioning model. This model is a robust multimodal system that combines a vision encoder with a large language model (LLM) to enable a comprehensive understanding of both visual and textual information.

For each patent, we experiment with different prompts to generate suitable captions. We observe that the patent title typically represents the name of the object, but lacks details about the shape or functionality of the design. As the final caption generation prompt, we use the following: *This is the image of {patent_title}. What is the shape of the image? What is the functionality of {patent_title}?* Figure 3 illustrates examples of several design patents, along with their titles and our generated captions. The examples show that the generated captions reflect the provided instructions. We also utilize these captions and demonstrate that they are beneficial for patent related tasks (see Sec. 4).

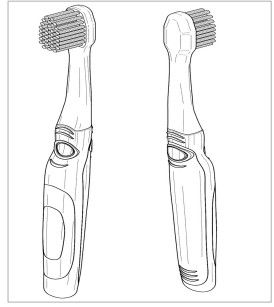 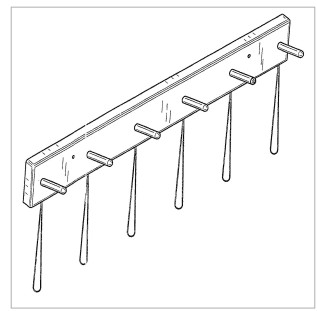 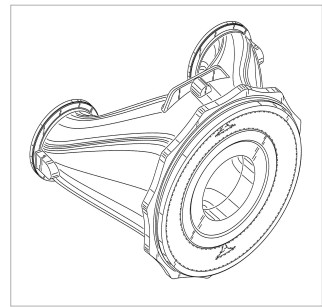

**Title: Oral care implement**
**Caption:** The image is a drawing of a toothbrush, which is an oral care implement used for cleaning teeth. The toothbrush has a long handle and a head with bristles, designed to effectively remove plaque and food particles from teeth and gums. The toothbrush is an essential tool for maintaining good oral hygiene and preventing dental issues such as cavities and gum disease.

**Title: Utility rack**
**Caption:** The image is a rectangle, and it features a wooden utility rack with several hooks. The functionality of the utility rack is to provide a convenient and organized storage solution for various items, such as tools, utensils, or small equipment. The hooks allow users to hang items on the rack, keeping them off the countertops and keeping the workspace tidy and clutter-free.

**Title: Portion of a socket**
**Caption:** The image is a black and white drawing of a cylindrical object with a hexagonal pattern. The object is a portion of a socket, which is a type of electrical connector that allows for the safe and secure attachment of electrical devices, such as light fixtures or appliances, to an electrical supply. Sockets are designed to provide a stable and secure connection between the device and the electrical supply, ensuring that the device can operate safely and efficiently.

Figure 3: It shows examples of three different design objects, titles and the generated captions from our dataset. The generated captions contain shapes and functionalities of the objects.

Table 1: Zero-shot multimodal retrieval results on captions generated using different models. GPT-4o performs the best and LLaVA outperforms Qwen-VL.

| Model | Image-Text | | Text-Image | |
|---|---|---|---|---|
| | R@5 | R@10 | R@5 | R@10 |
| Qwen-VL | 6.6 | 10.8 | 9.4 | 13.3 |
| GPT-4o | 10.5 | 15.3 | 12.4 | 17.5 |
| LLaVA | 8.4 | 12.2 | 9.5 | 13.8 |

**Potential bias and limitations of our IMPACT dataset.** First, IMPACT covers design patents filed between 2007 and 2022 within the US and documented in English. These temporal and geographical limitations exclude earlier design trends and patents from other regions. It might potentially impact the applicability of our findings across different temporal and global contexts. Second, the number of figures have a high variance across patents. On average, each patent consists of 7 to 9 figures, while some contain as few as 2-3 and others more than 20. This might become an issue if the downstream tasks depend on all figures (e.g., 3D image construction task) and it may not fully represent the variability in terms of the figures.

# 4 Patent Analysis with IMPACT

Our IMPACT dataset—on design patents—facilitates multimodal analysis tasks such as classification, retrieval, and similarity assessment. We perform classification and retrieval on our IMPACT dataset and demonstrate the usefulness of elaborated captions. Additional results are provided in the Appendix.

## 4.1 Classification

Patent reviewers usually are responsible to classify the patent applications, i.e., they assign the design codes. This is time-consuming due to the numerous classification codes. For instance, the U.S. design patent system has 33 classes which are further divided into subclasses. Given that design patents include both titles and visual content, the goal of the experiment is patent classification by integrating titles, captions, and images. Although a single design patent can be associated with multiple design codes, we focus on the primary classification—solving the task as a multi-class classification problem.

**Setup.** For this patent classification task, we use the recent patents from 2021 and obtain a total of 32,536 patents. We exclude the sparsely populated subclasses that appear only twice or less. Consequently, the final dataset has 25,500 patents with a total of 228084 images and they are categorized into 835 classification codes (subclasses).

We test five different combinations of input features: only title, only caption, title and image, caption and image, and a combination, i.e, concatenation of all three. All text-based features are extracted using RoBERTa [36], while image features are obtained via the image encoder of CLIP [41]. Afterwards, we employ a range of classification algorithms which include Support Vector Machines (SVM), Multinomial Naive Bayes (MultiNB), Logistic Regression (LR), and Multilayer Perceptrons (MLP). The primary objective here is to assess the power of images and texts (e.g., elaborated caption and title) for patent classification. We also applied BERT[10] on only text data (title and caption).

**Results.** Table 2 presents the classification average accuracy over all labels and F1 scores for all the combinations. BERT can be only applied on text-based features. It produces similar results with title and the generated caption. The best results for other individual classifiers are in bold. The results demonstrate that the best results—both accuracy and F1 scores— are produced by the combined features with title, caption, and image. The combine features outperform the features only with caption and image quite significantly in many cases and up to 36% in terms of accuracy. This also shows the importance of the generated captions in patent classification.

Table 2: Patent classification using various machine learning models. Combined denotes the combination of the features from the title, caption, and image. The best accuracy and F1-scores are shown in bold for individual methods. The combined features produce the best results consistently.

| Metrics | BERT | SVM | MultiNB | LR | MLP |
|---|---|---|---|---|---|
| **Acc. (Title)** | 0.55 | 0.55 | 0.38 | 0.54 | **0.52** |
| **Acc. (Caption)** | 0.52 | 0.46 | 0.38 | 0.45 | 0.41 |
| **Acc. (Title+Image)** | - | 0.58 | 0.44 | 0.57 | 0.51 |
| **Acc. (Caption+Image)** | - | 0.44 | 0.44 | 0.52 | 0.46 |
| **Acc. (Combined)** | - | **0.60** | **0.51** | **0.59** | **0.52** |
| **F1 Score (Title)** | 0.53 | 0.53 | 0.33 | 0.52 | 0.49 |
| **F1 Score (Caption)** | 0.50 | 0.37 | 0.33 | 0.44 | 0.39 |
| **F1 Score (Title+Image)** | - | 0.57 | 0.41 | 0.56 | 0.50 |
| **F1 Score (Caption+Image)** | - | 0.51 | 0.41 | 0.50 | 0.45 |
| **F1 Score (Combined)** | - | **0.58** | **0.49** | **0.57** | **0.50** |

## 4.2 Multimodal Retrieval

The patent retrieval task is to identify relevant patent documents and images in response to search queries. This process is often used for discovering new patents and assessing their novelty. However, recent patent retrieval systems mainly work on retrieving images only from query images. Here, we focus on multimodal retrieval, which incorporates both text and images. This integration enhances the ability to cross-reference and verify information, thus improving the overall effectiveness and

efficiency of patent searches. Additionally, multimodal retrieval can enable creativity and innovation in design by providing richer, more diverse sources of inspiration. We conduct experiments on text-image (T2I) and image-text (I2T) retrieval tasks.

**Setup.** In this task, we use the patents from recent five years. We randomly shuffle patents in different years and split into train/val set. The training set has 113,887 patents, and the validation set has 5,000 patents. For each patent, we construct an image-title pair and an image-caption pair for ablation studies. We also utilize our entire dataset in some experiments. Note that, we describe the caption generation process in Section 3. We conduct T2I and I2T retrieval tasks on both zero-shot and finetuned setups. We use CLIP [41] to train and inference from patent images (captions) and their corresponding captions (images) in this task. Specifically, in T2I retrieval, given a title or a caption, we measure whether the ground truth image is within the top K retrieved results. We evaluate on validation sets to compare the performance of using image-title and image-caption pairs.

*PatentCLIP:* We also provide PatentCLIP models, which are finetuned for 30 epochs from pre-trained CLIP. PatentCLIP is finetuned on over 113k patent image-caption pairs on ITM[7] for all backbones. The backbone models are ResNet [15] and ViT [11] families. More details on training and hyperparameters are in Appendix A.4. All experiments are conducted on a node of 4 NVIDIA V100 GPUs. Additionally, we provide image retrieval results using PatentCLIP and comparison with other state of the art patent models in Table 4b.

Table 3: Multimodal retrieval tasks in zero-shot and finetuned settings. Dataset denotes the image-text pairs used in the experiments. The best Recall@K (%) are shown in bold. Image-caption pairs produce the best results consistently.

|  | Dataset | Backbone | Text-Image | | | Image-Text | | |
|---|---|---|---|---|---|---|---|---|
|  |  |  | R@1 | R@5 | R@10 | R@1 | R@5 | R@10 |
| Zero-shot | Image-Title | ResNet50 | 0.52 | 2.10 | 3.32 | 0.20 | 0.72 | 1.64 |
|  |  | ResNet101 | 1.02 | 3.20 | 4.72 | 0.30 | 0.82 | 1.28 |
|  |  | ViT-B-32 | 1.06 | 3.54 | 5.56 | 0.38 | 1.62 | 2.60 |
|  |  | ViT-L-14 | 2.78 | 7.38 | 10.40 | 1.16 | 4.30 | 7.32 |
|  | Image-Caption | ResNet50 | 0.82 | 2.52 | 4.08 | 0.78 | 2.32 | 3.48 |
|  |  | ResNet101 | 1.44 | 4.52 | 6.48 | 0.98 | 2.98 | 4.96 |
|  |  | ViT-B-32 | 1.98 | 5.24 | 7.42 | 1.06 | 4.26 | 6.32 |
|  |  | ViT-L-14 | **4.46** | **10.74** | **15.16** | **3.42** | **8.90** | **12.88** |
| Finetuned | Image-Caption | ResNet50 | 5.38 | 15.52 | 22.7 | 5.9 | 16.6 | 23.86 |
|  |  | ResNet101 | 7.44 | 20.6 | 28.48 | 7.02 | 19.70 | 27.58 |
|  |  | ViT-B-32 | 10.24 | 25.56 | 35.06 | 9.88 | 25.90 | 35.08 |
|  |  | ViT-L-14 | **20.58** | **43.14** | **53.00** | **20.44** | **42.34** | **52.56** |

Table 4: Results on Image to Text, Text to Image, and Image Retrieval Tasks.

| Model | Image-Text | | Text-Image | |
|---|---|---|---|---|
|  | R@5 | R@10 | R@5 | R@10 |
| ResNet50 | 23.92 | 32.84 | 25.00 | 34.37 |
| ResNet101 | 25.82 | 35.25 | 26.60 | 36.06 |
| ViT-B-32 | 28.41 | 38.32 | 29.28 | 39.87 |
| ViT-L-14 | 39.59 | 50.44 | 41.72 | 52.55 |

(a) Multimodal retrieval tasks in finetuned settings on entire dataset.

| Model | mAP |
|---|---|
| ViT-B + ArcFace [17] | 0.614 |
| CLIP-ViT-B + ArcFace | 0.645 |
| **PatentCLIP-ViT-B + ArcFace** | **0.657** |

(b) Image retrieval performance comparison with other models.

**Results.** We measure multimodal retrieval performance (Recall@K[8]) in zero-shot and finetuned setups. Table 3 shows that performance in retrieval with image-caption pairs outperforms the same with image-title pairs in all settings. We also provide finetuned results on image-title dataset in Appendix A.5. As the results shown, finetuned models gain significant improvements, by up to *550%*

---

[7]Contrastive Image-text matching training objective

[8]The metric R@K evaluates whether the ground truth appears within the top K results of the validation set.

and *830%* respectively at R@1 on T2I and I2T retrievals. Note that, ViT-L obtains the best recall in all settings, which demonstrates that the larger advanced models boost the performance. We also measure the multimodal retrieval performance on the entire dataset. Since Image-Caption pair give us the highest recall (See Table 3), we use this pair for the task shown in Table 4a. Table 4b shows that CLIP, when fine-tuned on our dataset, outperforms other models on image retrieval tasks and achieves the highest mAP. These results indicate that more descriptive training data, such as detailed captions versus simpler titles, can effectively improve the models' performance. Moreover, finetuning allows the model to adjust its parameters better aligned with the patent domain. Thus, we believe that a scaled patent multimodal foundation model is beneficial for analyses of design patents.

## 4.3 Human Evaluation on Generated captions

We conduct human evaluation to assess the generated captions from LLaVA. In the study, we have a total of 12 participants. All participants are graduate students in STEM with prior research experience. Each participant reviews three sets of titles, captions, and images. We ask the user to read these carefully and determine if the captions can add value to the image as a descriptive caption in their opinion with the following questions: *Did the caption describe the image correctly? Did the caption can describe the shape and functionality of the image logically?* As a result, in more than 60% of their responses, they agree with the quality of the generated captions. However, one of the examples shows the limitations of general multimodal captioning models and we would consider this as a future direction. Figure 4 shows three examples of design patents, generated captions, and human evaluation results.

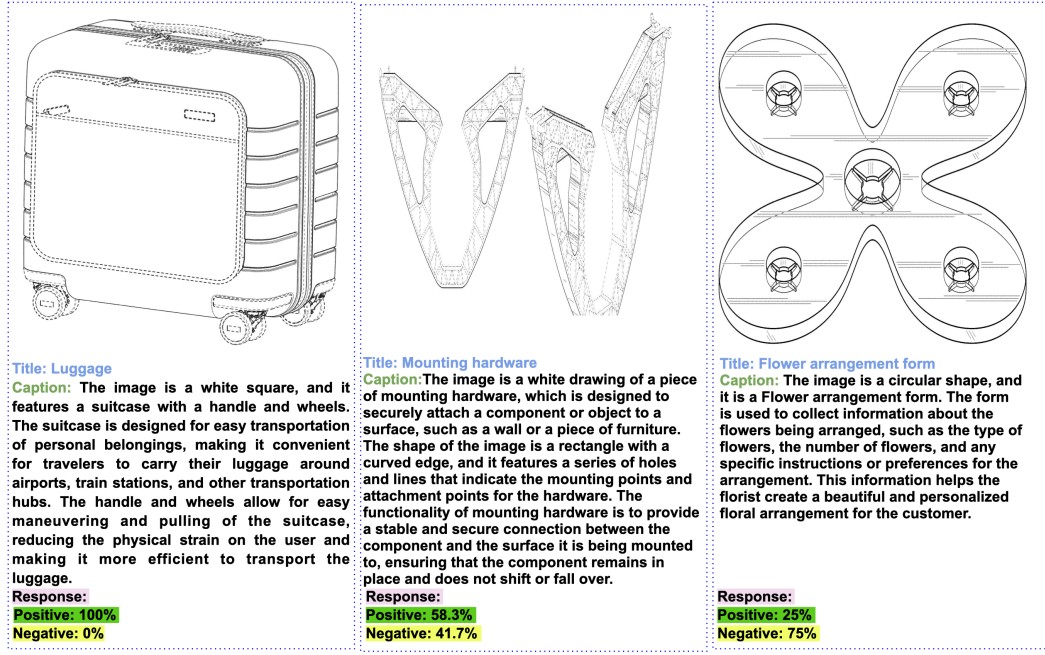

Figure 4: Examples of three different design objects, titles, generated captions, and human evaluation results on the generated captions. Positive means the caption describes the image, shape and functionality correctly and logically. Negatives means the opposite.

## 4.4 Other Applications of IMPACT in Computer Vision

### 4.4.1 3D image reconstruction

3D image reconstruction is the process of creating a three-dimensional model from one or more two-dimensional images. The ability to produce rapid and precise 3D image reconstructions has applications in numerous CV fields, including medical, robotics, entertainment, reverse engineering, augmented reality, human-computer interaction, and animation [24]. Recent studies have focused on the reconstruction of three-dimensional images from two-dimensional images [49, 52, 38]. Most

existing research in the field of 3D image reconstruction from 2D images use the standard ImageNet dataset as a benchmark [9].

*Our* IMPACT *dataset.* The generic approach involves using an image generator to produce a tri-plane representation with image depths. Multiple viewpoints helps the process as it provides the information about different angles and accurate depth estimation. The advantage of IMPACT is that is consists of design sketches from many viewpoints for the same design and that can be used to generate 3D models. Thus, IMPACT can serve as a benchmark for 3D image construction and as a valuable source of design inspiration for future inventors. It can also help the reviewers to search the prior art to mitigate infringement as 3D constructions are more informative than a 2D model. We provide examples in Appendix A.6.

### 4.4.2 Visual Question Answering

Visual Question Answering (VQA) asks to predict answer classes or generate a short phrase for the given images and textual questions [3]. VQA enhances machine understanding of visual content and enables various practical applications across different industries, such as Medical VQA [16].

*Our* IMPACT *dataset.* We propose a new task PatentVQA, which offers significant advantages in understanding design patents. It can enhance interpretation by simplifying complex visual data and automates the analysis of visual patent information, thus it can help in speeding up the review process. First, we propose to construct and annotate an IMPACT-VQA dataset. Then, one can develop a VQA systems using Vision-Language models [35, 31], which makes patent databases more accessible to a broader audience and reduces the need for specialized training and helping users engage more deeply with patent content. We illustrate an example in Fig. 1 and additional examples are in Appendix A.7. Therefore, we believe that PatentVQA is a valuable tool in patent analysis and design innovation.

## 5 Conclusions

In this work, we have developed a large dataset on design patents, IMPACT that integrates visual and textual information within a unified framework. We have compiled a collection of 435,101 design patents spanning 16 years with 3.6 million drawing figures along with eleven fields. IMPACT is a multimodal patent dataset with patent images and detailed captions to support various computer vision and multimodal tasks. IMPACT is particularly valuable for advanced patent analysis tasks such as classification, retrieval, prior art searches, and design trend analysis. We demonstrate the usefulness of IMPACT in two specific tasks: classification and retrieval. The results demonstrate that the integration of additional textual information along with the existing images help in improving performance in several settings.

**Broader Impact.** Patent data—even for design patents—is publicly available, but there has been a need to organize this large amount of data for the recent deep learning methods and applications. We believe IMPACT will serve as an important benchmark to both patent and computer vision community. This work is about creating an organized dataset from publicly available data and we do not foresee any potential negative societal impact of it.

**Future Directions.** Besides the applications proposed earlier, another interesting future direction could be to develop efficient foundation models specifically for patent data. These can significantly enhance the accuracy and relevance of patent analyses, as generic vision-language models often underperform in this type of specialized domain.

# 6 Acknowledgments

We thank the reviewers for their valuable suggestions and 12 participants for human evaluations. This work was supported by in-kind contributions from University of Illinois Urbana-Champaign (UIUC) HACC Cluster and NSF ACCESS UIUC NCSA Cluster (Ref: ELE230014). Zhu Wang was supported by Sathya N. Ravi's UIC start-up funds.

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

# A   Appendix

## A.1   Additional Details on Our Dataset

**Number of words in a title.** Design patents typically contain limited textual information. One of the fields is the title, which describes the object the design patent protects. Figure 5 shows the number of words in each title over the years. It is noteworthy that the titles average only about 3 to 3.5 words, indicating a trend towards concise and specific naming conventions in design patent filings.

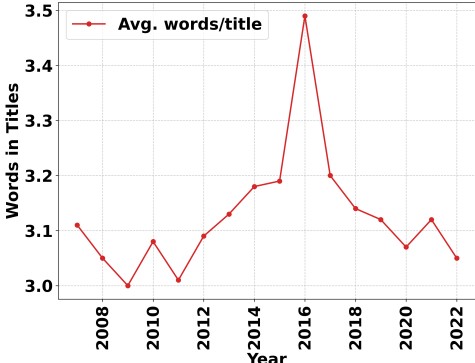

**Figure 5:** The distribution of the number of words in the title of the design patents over the years. This shows that the average number of words in the title of each patent is between 3 and 3.5.

**Frequent objects.** Figure 6 displays the top 10 objects that occur the most frequently in the data set. Display screens are the most common, with a frequency of 13,810, followed by shoes and bottles with frequencies of 4,327 and 3,718, respectively. Other frequently appearing objects include containers, mobile phones, chairs, tires, shoe uppers, electric devices, and faucets.

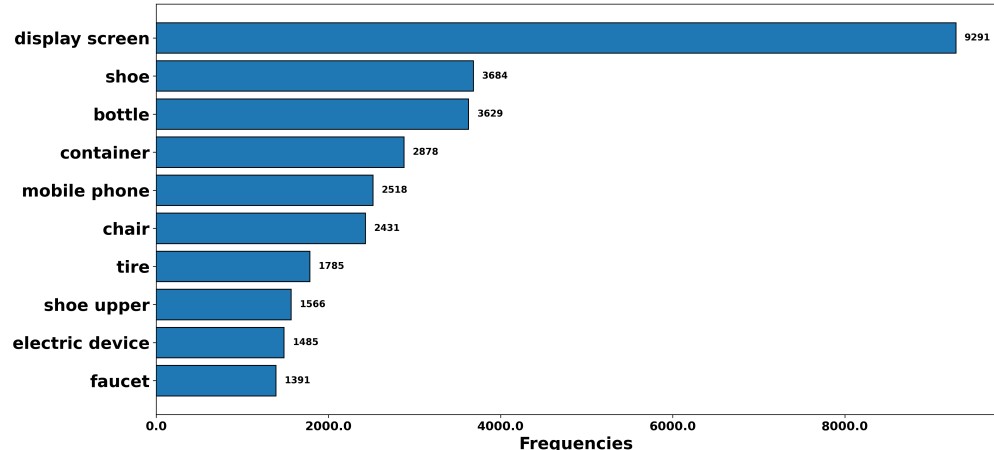

Figure 6: Distribution of the top 10 objects in the dataset by frequency. Display screens, including different types such as graphical user interfaces, animated graphical user interfaces, and transitional graphical user interfaces, are the most common objects in the dataset.

**Example of the fields.** In the constructed CSV files, we have 11 fields for each design patent. Table 5 shows the descriptions of the 11 fields extracted from the XML files with an example. The description defines the meaning of each column, and the examples of the fields are provided for the patent ID D0908314.

Table 5: Description of fields in the CSV file. There are 11 fields extracted from the XML files for each design patent. The description defines the meaning of each column, and the examples of the fields are provided for the patent ID D0908314.

| Column | Description | Example |
|---|---|---|
| title | The title of the design is the name commonly recognized and used by the public | Garment with a side pocket |
| id | Document number of the patent starts with a 'D' followed by a series of numbers, which uniquely identifies the patent. | D0908314 |
| claim | A design patent application includes only a single claim that defines the design the applicant wishes to patent, specifying the article in which the design is embodied or to which it is applied. | The ornamental design for a garment with a side pocket, as shown and described. |
| date | Publication date of the patent | 20210126 |
| class | U.S. design patent category under which the patent is classified | D2728, D2840 |
| class_search | U.S. classification codes aiding in determining its scope and relevant prior art | ['0202', 'D2728', 'D2839', 'D2829', 'D2750', 'D2839', 'D 2750', 'D2857', 'D2840', 'D2829', 'D 2840', 'D2840', 'D2840', 'D2829', 'D 2839', 'D2840', 'D21804', '293', '224153', 'D2712', 'D2720', 'D2750', 'D2831', 'D2853', 'D2865', 'D2873', 'D2874', 'D2878', 'D2840', 'D2839', 'D2857', 'D2728', 'D21801-805'] |
| inv_country | Country of the inventors | US |
| no_figs | Number of figures for the design | 7 |
| sheets | Number of design sheets provided for the figures. Some of the sheets has multiple figure views | 4 |
| file_names | The filenames that contain the images of the particular design. | ['USD0908314-20210126-D00000.TIF', 'USD0908314-20210126-D00001.TIF', 'USD0908314-20210126-D00002.TIF', 'USD0908314-20210126-D00003.TIF', 'USD0908314-20210126-D00004.TIF'] |
| fig_desc | The Figure Descriptions specify the representation of each drawing view, such as front view, top view, perspective, and others | ['FIG. 1 is a front left perspective view of the garment with a side pocket, showing my new design', 'FIG. 2 is a front view', 'FIG. 3 is a rear view', 'FIG. 4 is a left side view', 'FIG. 5 is a right side view', 'FIG. 6 is a top view thereof', 'FIG. 7 is a bottom view'] |
| caption | Elaborated captions for the design which includes shape and functionality | The image is a square-shaped illustration of a garment with a side pocket. The functionality of the image is to showcase the design and features of the garment, such as the pocket, which can be useful for potential customers or designers to visualize the product and its details. |

**Frequent USPC class.** Table 6 lists the top 10 most frequent class-subclass occurrences in the dataset. The most common class-subclass is D14-486, "Recording, Communication, or Information Retrieval Equipment," with 7,618 occurrences, specifically describing drop-down or full-screen menu types. Other notable entries include D14-485, D26-28, and D12-209, covering generated images, vehicle lamps or casings, and transportation apertures, respectively.

Table 6: Overview of the 10 Class-Subclass Occurrences. This lists the frequency of occurrences for each class-subclass, along with descriptions that specify the general category and particular functionalities or features characterized by each subclass. Note that some patents belong to multiple classes. For simplicity, we have counted only the primary class and that results in a single classification code for each patent.

| Class-Subclass | Occurrence | Class Description | Subclass Description |
|---|---|---|---|
| D14-486 | 7359 | Recording, Communication, or Information Retrieval Equipment | Drop down menu or full screen menu type |
| D14-485 | 5363 | Recording, Communication, or Information Retrieval Equipment | Generated image |
| D26-28 | 4149 | Lighting | Vehicle lamp or casing |
| D12-209 | 2700 | Transportation | Aperture or simulated aperture |
| D2-972 | 2432 | Apparel and Haberdashery | Vamp, toe, heel, or side panel |
| D12-169 | 2399 | Transportation | Vehicle-attached front or rear type |
| D14-250 | 2320 | Recording, Communication, or Information Retrieval Equipment | Cover for base or handset |
| D13-147 | 2201 | Equipment for Production, Distribution, or Transformation of Energy | Linear array of identical repeating ports or contacts (i.e., in-line array) |
| D14-488 | 2022 | Recording, Communication, or Information Retrieval Equipment | Visible shutter |
| D14-126 | 1998 | Recording, Communication, or Information Retrieval Equipment | Receiver or monitor |

**Example viewpoints.** We provide examples of the drawings that are available in the design patents. Figure 7 shows four viewpoints of a design object titled as *ceiling fan*. 7a and 7b represent top and bottom perspectives respectively, whereas 7c and 7d are top and bottom view, respectively.

## A.2 Generated Captions Examples and Comparisons

For caption generation, we use LLaVA, GPT-4o, and Qwen-VL. Qwen-VL often lacks functional descriptions and occasionally includes Chinese words within English captions. Although GPT-4o provides high-quality captions, it comes with a higher cost. Figure 8 illustrates an example of a design patent along with the captions generated by each model.

## A.3 US Design Patent Classification

The subject matter of U.S. design patents is categorized into 33 distinct classes[9]. Unlike utility patents, design patents are organized by classes and subclasses only. Table 7 lists all the categories of the design patents.

## A.4 PatentCLIP

We also provide a PatentCLIP which is finetuned from OpenAI's CLIP pre-trained models with IMPACT dataset. In this section, we illustrate the implementation details and qualitative analysis using the learned feature embedding space from IMPACT.

---

[9]https://www.uspto.gov/patents/laws/examination-policy/seven-classification-design-patents

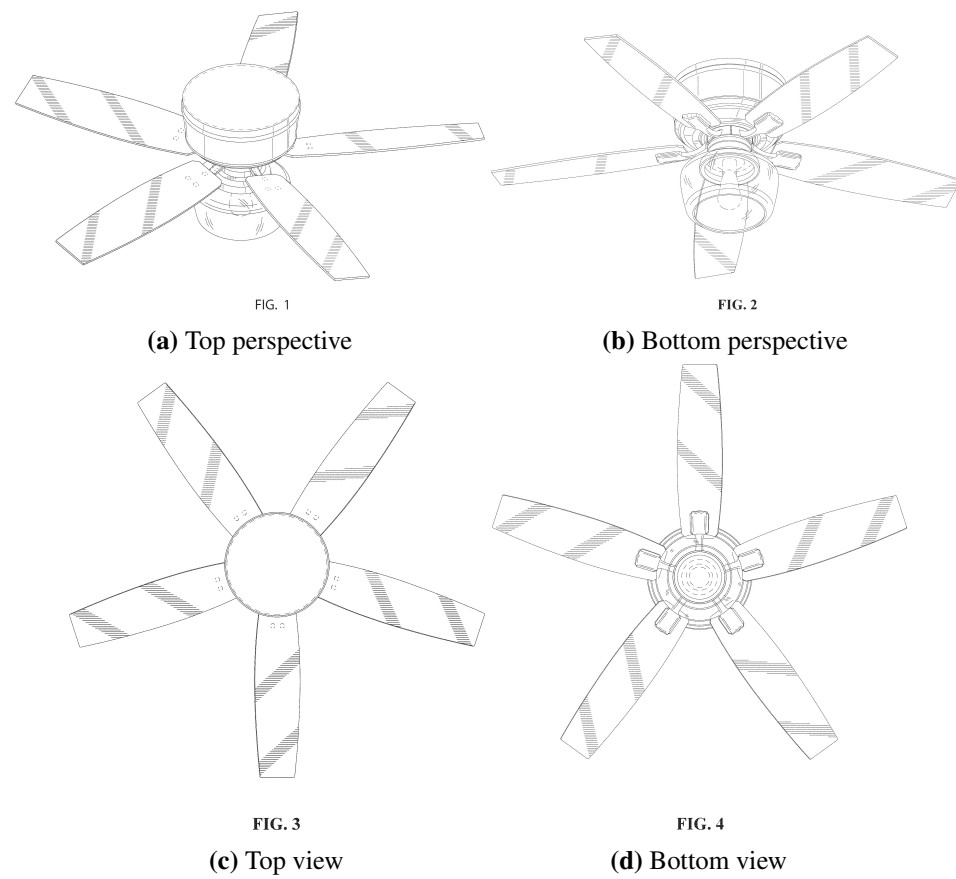

**(a)** Top perspective      **(b)** Bottom perspective

**(c)** Top view      **(d)** Bottom view

Figure 7: An example of image perspectives and viewpoints in a design patent. The patent is about the design of a ceiling fan.

### A.4.1 Implementation details

We use an open source implementation of CLIP [10]. The models are ResNet50, ResNet101, ViT-B-32 and ViT-L-14. The hyperparameters for funetining is listed as follows: learning rate is $5e - 6$, weight decay is 0.1, and optimizer is AdamW for all models. The batch size is 256 except 64 for ViT-L-14. All settings are same to image-title and image-caption pairs. All finetuning and inference are conducted on 4 NVIDIA V100 GPUs.

### A.4.2 Qualitative analysis

To analysis the effective of finetuning CLIP on IMPACT dataset, we visualize the learned image features and text features for sample patents using U-MAP projection [37]. The sample patents are selected from the top 4 subclass in the recent five years data, including D12-209, D14-485, D14-486 and D26-28. In total, there are 5,699 patents. All the model backbone is ViT-B-32. For text features, we visualize the embeddings for the captions.

Feature embedding spaces of multiple modalities on sample IMPACT dataset are shown in Fig 9. Different colors representing the clusters of the corresponding classes. We observe that PatentCLIP can identify clusters over the extracted image features better than CLIP, see 9a, 9c and 9e. Comparing with PatentCLIP-title, PatentCLIP have the better clustering performance on the extracted text features, it can identify D14 is far to D12. Note that, CLIP also can identify text feature clusters because that the captions are generated with VLMs. However, CLIP is not able to classify the patent images. Indeed, we can see that different classes cluster clearly, and similar subclasses are often close in the embedding spaces, such as D14-485 and D14-486 (see class description in Tab 6). Therefore,

---

[10]https://github.com/mlfoundations/open_clip

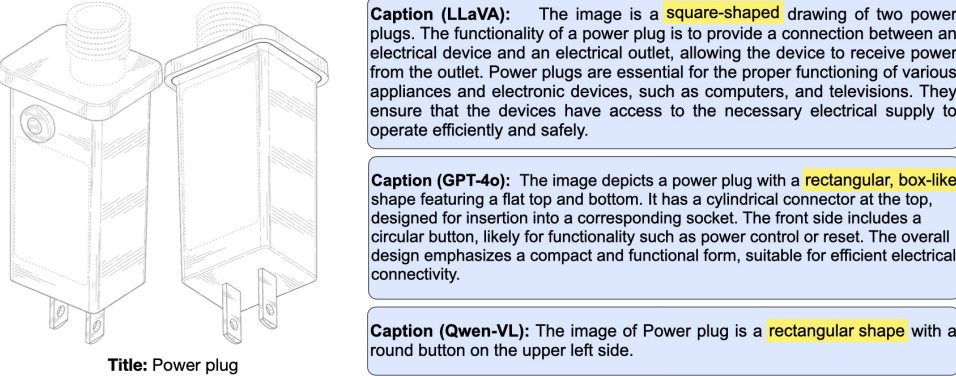

**Caption (LLaVA):** The image is a square-shaped drawing of two power plugs. The functionality of a power plug is to provide a connection between an electrical device and an electrical outlet, allowing the device to receive power from the outlet. Power plugs are essential for the proper functioning of various appliances and electronic devices, such as computers, and televisions. They ensure that the devices have access to the necessary electrical supply to operate efficiently and safely.

**Caption (GPT-4o):** The image depicts a power plug with a rectangular, box-like shape featuring a flat top and bottom. It has a cylindrical connector at the top, designed for insertion into a corresponding socket. The front side includes a circular button, likely for functionality such as power control or reset. The overall design emphasizes a compact and functional form, suitable for efficient electrical connectivity.

**Caption (Qwen-VL):** The image of Power plug is a rectangular shape with a round button on the upper left side.

**Title:** Power plug

**Figure 8:** Captions generated by LLaVA, GPT-4o, and Qwen-VL of a power plug design. The shapes are highlighted in yellow.

we believe that finetuning VLMs with IMPACT is beneficial in the specific patent domain for many downstream tasks, such as classification and retrieval.

### A.5 Multimodal Retrieval

#### A.5.1 Additional results

We provide additional results on the performance of finetuned PatentCLIP-title on Text-Image and Image-Text retrievals. All the training hyper-parameters are same as the PatentCLIP model (see Sec A.4). The dataset used here is the image and title pairs in IMPACT. As results shown in Table 8, the performance pattern is similar to PatentCLIP, which is that the more advanced models and the models with more parameters perform better. Comparing the results of PatentCLIP which is finetuned on image and caption pairs in Table 3 is similar except for R@1 results of ResNet family. Others R@K results of finetuning image-captions are significantly improved than PatentCLIP-title.

#### A.5.2 Qualitative analysis

To further analyze the multimodal retrieval results, we demonstrate three Text-image retrieval examples as follows.

- Example 1: *Text Query:* The image is a square-shaped drawing of a protective case for a game controller. *Ground truth image:* D1006114.TIF
- Example 2: *Text Query:* The image is a black and white drawing of a computer mouse, which is a device used for controlling and interacting with a computer. *Ground truth image:* D0943581.TIF
- Example 3: *Text Query:* The image is a white drawing of an FM transmitter, which is a device used to transmit audio signals through the air using frequency modulation (FM) technology. *Ground truth image:* D0985524.TIF

Figure 10 shows that PatentCLIP-title and PatentCLIP are able to retrieval game controller, but CLIP only can recall the items with square-shaped. All top 5 retrieved images of PatentCLIP are related with game controller, and the top 1 result is correct. As results shown in Fig 11, both CLIP and PatentCLIP obtained the correct images in top 5 set, but PatentCLIP produce the top 1 image correctly. Other four images and the images obtained from PatentCLIP-title are not relevant with the text query. In Figure 12, only PatentCLIP retrieve the top 1 image correctly. We see that the retrieved images by our PatentCLIP model are relevant to the text queries not only in the shape but also in terms of the function. Thus, we conclude that captions provide more information for VLMs to learn the patents. Our PatentCLIP model is also helpful for the prior art search and the design inspiration.

### A.6 3D construction Examples

We provide two more detailed 3D constructions examples in Fig. 13. We utilize ControlNet [58] to generate 3D photos for patent images. Comparing the results of prompting with IMPACT captions

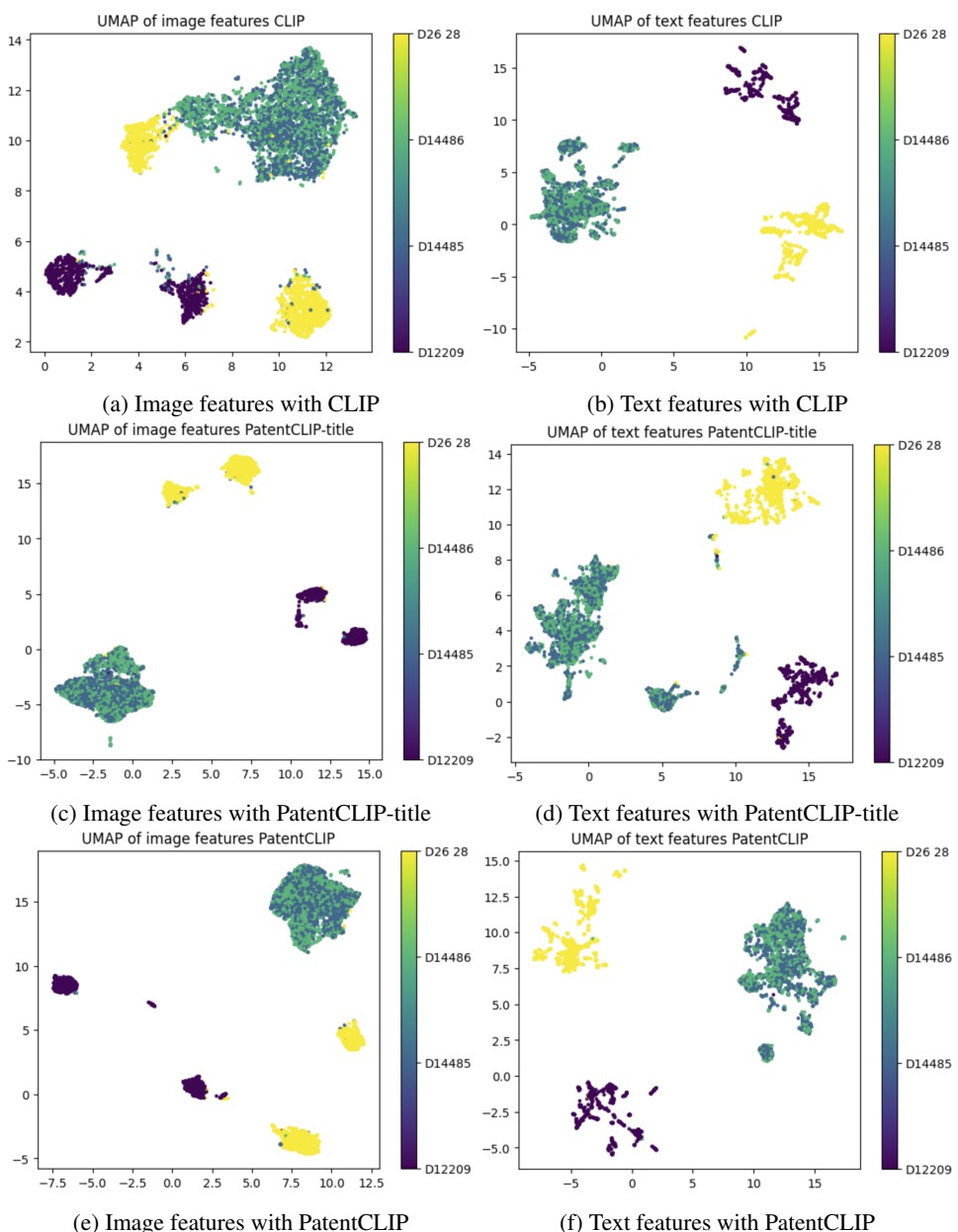

Figure 9: UMAP feature embeddings for sample patent images. (a) Visualization using CLIP models (b) Visualization using PatentCLIP finetuned on title (c) Visualization using PatentCLIP finetuned on caption. PatentCLIP shows well formed clusters in both image and text-based features.

Table 7: The table shows the list of U.S. design patent classes.

| Class | Description |
|---|---|
| D1 | Edible Products |
| D2 | Apparel and Haberdashery |
| D3 | Travel Goods, Personal Belongings, and Storage or Carrying Articles |
| D4 | Brushware |
| D5 | Textile or Paper Yard Goods; Sheet Material |
| D6 | Furnishings |
| D7 | Equipment for Preparing or Serving Food or Drink Not Elsewhere Specified |
| D8 | Tools and Hardware |
| D9 | Packages and Containers for Goods |
| D10 | Measuring, Testing or Signaling Instruments |
| D11 | Jewelry, Symbolic Insignia, and Ornaments |
| D12 | Transportation |
| D13 | Equipment for Production, Distribution, or Transformation of Energy |
| D14 | Recording, Communication, or Information Retrieval Equipment |
| D15 | Machines Not Elsewhere Specified |
| D16 | Photography and Optical Equipment |
| D17 | Musical Instruments |
| D18 | Printing and Office Machinery |
| D19 | Office Supplies; Artists' and Teachers' Materials |
| D20 | Sales and Advertising Equipment |
| D21 | Games, Toys and Sports Goods |
| D22 | Arms, Pyrotechnics, Hunting and Fishing Equipment |
| D23 | Environmental Heating and Cooling, Fluid Handling and Sanitary Equipment |
| D24 | Medical and Laboratory Equipment |
| D25 | Building Units and Construction Elements |
| D26 | Lighting |
| D27 | Tobacco and Smokers' Supplies |
| D28 | Cosmetic Products and Toilet Articles |
| D29 | Equipment for Safety, Protection and Rescue |
| D30 | Animal Husbandry |
| D32 | Washing, Cleaning or Drying Machines |
| D34 | Material or Article Handling Equipment |
| D99 | Miscellaneous |

Table 8: Multimodal retrieval tasks in finetuned settings for image-text pairs used in the experiments. The best Recall@K (%) are shown in bold.

| | Dataset | Backbone | Text-Image | | | Image-Text | | |
|---|---|---|---|---|---|---|---|---|
| | | | R@1 | R@5 | R@10 | R@1 | R@5 | R@10 |
| Finetuned | Image-Title | ResNet50 | 5.44 | 14.98 | 21.54 | 5.18 | 14.46 | 20.28 |
| | | ResNet101 | 7.76 | 18.98 | 24.66 | 7.16 | 18.30 | 24.38 |
| | | ViT-B-32 | 9.16 | 22.38 | 29.20 | 8.42 | 22.18 | 28.86 |
| | | ViT-L-14 | **14.88** | **33.04** | **41.42** | **13.75** | **31.98** | **39.96** |

and prompting with patent title, we observe that our captions can provide more guidance for diffusion models.

## A.7 Visual Question Answering Examples

We provide two more detailed Visual question answering (VQA )examples. Based on IMPACT dataset, we design a set of questions which are relevant to design patents. Figures 14 and 15 are two examples with 3 questions and answers for each exampe patent images. We use LLaVa [35] to generate answers.

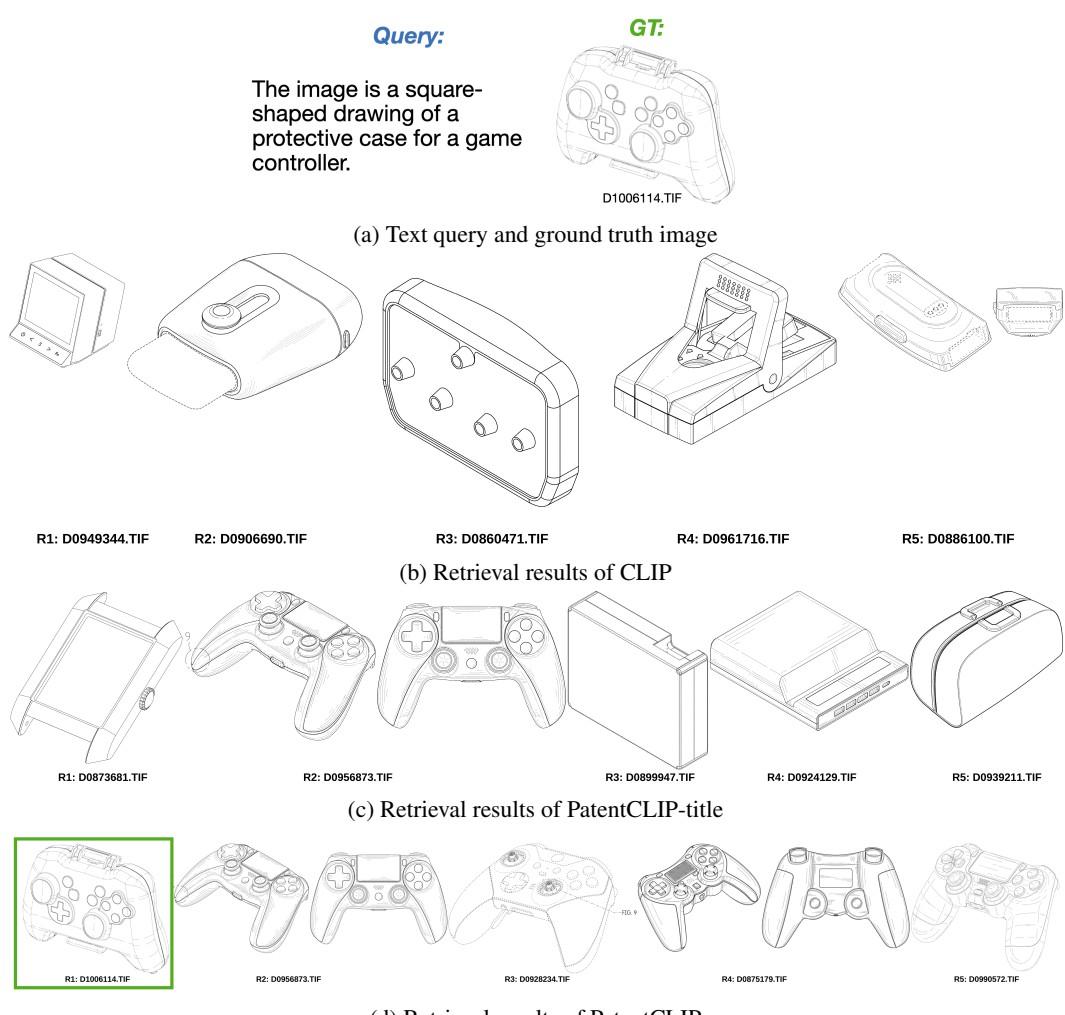

(a) Text query and ground truth image

(b) Retrieval results of CLIP

(c) Retrieval results of PatentCLIP-title

(d) Retrieval results of PatentCLIP

Figure 10: Text-image Retrieval example 1. Text query is shown in (a). (b), (c), and (d) are top 5 retrieval results of CLIP, PatentCLIP-title and PatentCLIP respectfully. Top 1-5 is from left to right. Green box denotes to the correct image. In this case, only PatentCLIP retrieves correctly, which means PatentCLIP learned relevant multimodal features in the patent domain.

We can see that the answers are general text, but they are still helpful for further patent analyses. Thus, we propose a IMPACT-VQA to be a patent specific domain, which can provide more patent related information VQA systems and will be a good future direction to explore.

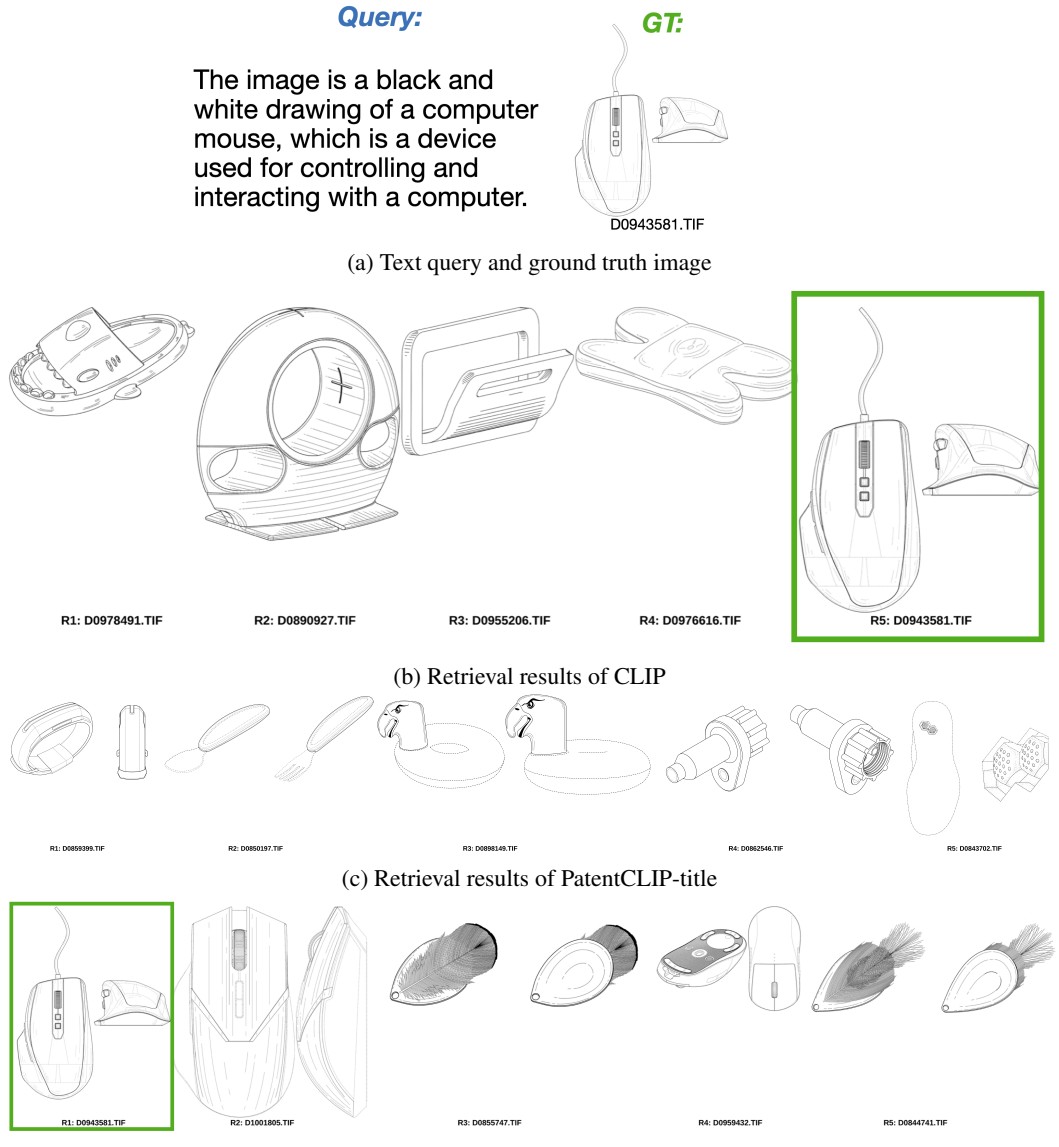

(a) Text query and ground truth image

(b) Retrieval results of CLIP

(c) Retrieval results of PatentCLIP-title

(d) Retrieval results of PatentCLIP

Figure 11: Text-image Retrieval example 2. Text query is shown in (a). (b), (c), and (d) are top 5 retrieval results of CLIP, PatentCLIP-title and PatentCLIP respectfully. Top 1-5 is from left to right. Green box denotes to the correct image. In this case, only CLIP and PatentCLIP retrieves correctly.

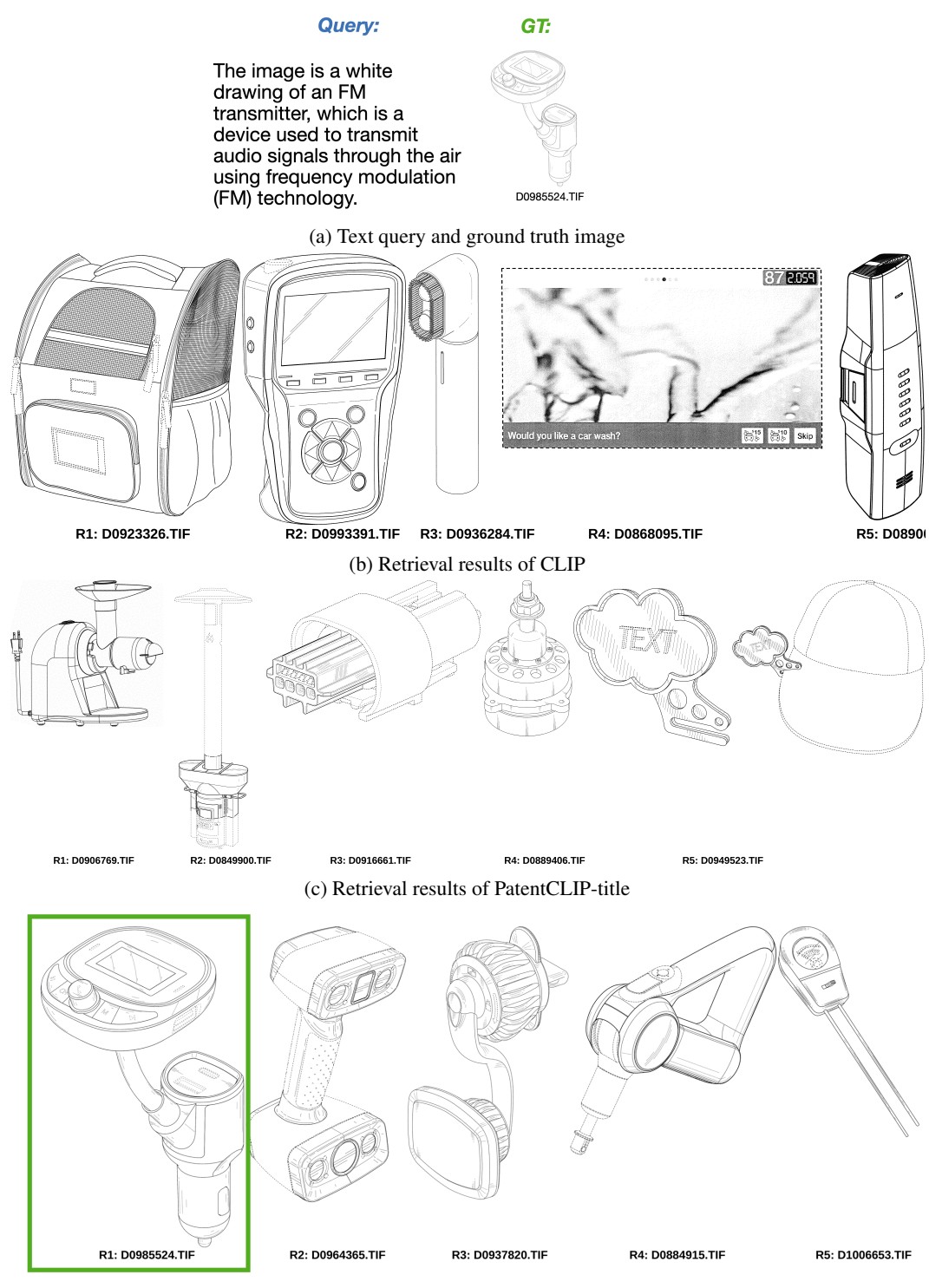

Query:

The image is a white drawing of an FM transmitter, which is a device used to transmit audio signals through the air using frequency modulation (FM) technology.

GT:

D0985524.TIF

(a) Text query and ground truth image

R1: D0923326.TIF    R2: D0993391.TIF    R3: D0936284.TIF    R4: D0868095.TIF    R5: D0890(

(b) Retrieval results of CLIP

R1: D0906769.TIF    R2: D0849900.TIF    R3: D0916661.TIF    R4: D0889406.TIF    R5: D0949523.TIF

(c) Retrieval results of PatentCLIP-title

R1: D0985524.TIF    R2: D0964365.TIF    R3: D0937820.TIF    R4: D0884915.TIF    R5: D1006653.TIF

(d) Retrieval results of PatentCLIP

Figure 12: Text-image Retrieval example 3. Text query is shown in (a). (b), (c), and (d) are top 5 retrieval results of CLIP, PatentCLIP-title and PatentCLIP respectfully. Top 1-5 is from left to right. Green box denotes to the correct image. In this case, only PatentCLIP retrieves correctly.

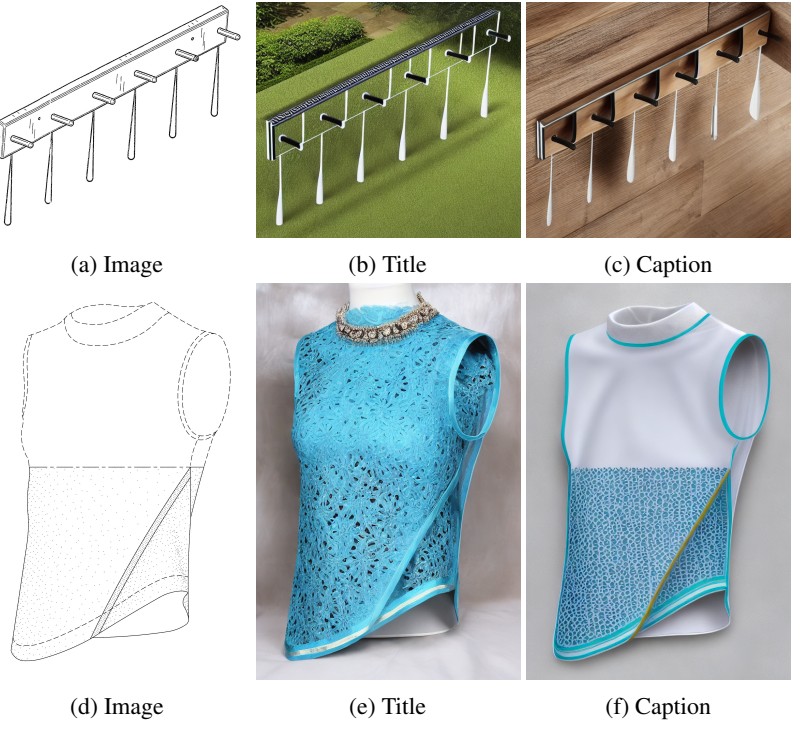

(a) Image      (b) Title      (c) Caption

(d) Image      (e) Title      (f) Caption

Figure 13: Examples of 3D reconstruction. (a) and (d) are patent images. (b) and (e) are generated images with title as text prompt. (c) and (f) are generated images with IMPACT captions.

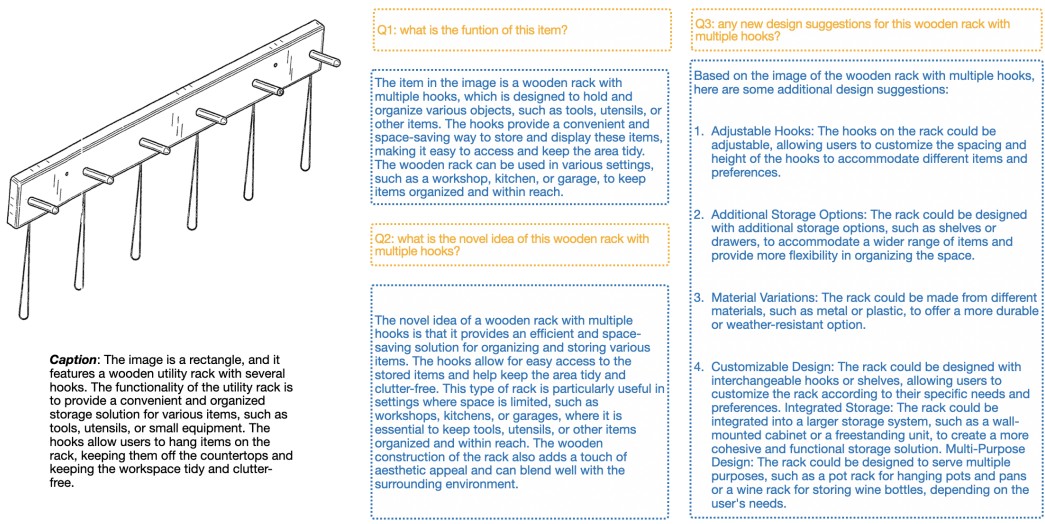

Figure 14: Detailed VQA example 1, given an patent image from IMPACT, we design a few questions and use LLaVA to generate answers.

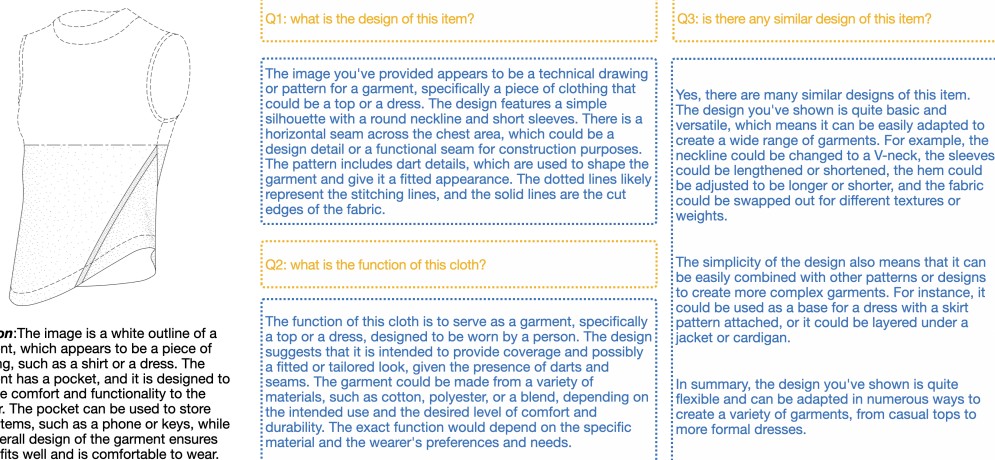

**Caption**: The image is a white outline of a garment, which appears to be a piece of clothing, such as a shirt or a dress. The garment has a pocket, and it is designed to provide comfort and functionality to the wearer. The pocket can be used to store small items, such as a phone or keys, while the overall design of the garment ensures that it fits well and is comfortable to wear.

Q1: what is the design of this item?

The image you've provided appears to be a technical drawing or pattern for a garment, specifically a piece of clothing that could be a top or a dress. The design features a simple silhouette with a round neckline and short sleeves. There is a horizontal seam across the chest area, which could be a design detail or a functional seam for construction purposes. The pattern includes dart details, which are used to shape the garment and give it a fitted appearance. The dotted lines likely represent the stitching lines, and the solid lines are the cut edges of the fabric.

Q2: what is the function of this cloth?

The function of this cloth is to serve as a garment, specifically a top or a dress, designed to be worn by a person. The design suggests that it is intended to provide coverage and possibly a fitted or tailored look, given the presence of darts and seams. The garment could be made from a variety of materials, such as cotton, polyester, or a blend, depending on the intended use and the desired level of comfort and durability. The exact function would depend on the specific material and the wearer's preferences and needs.

Q3: is there any similar design of this item?

Yes, there are many similar designs of this item. The design you've shown is quite basic and versatile, which means it can be easily adapted to create a wide range of garments. For example, the neckline could be changed to a V-neck, the sleeves could be lengthened or shortened, the hem could be adjusted to be longer or shorter, and the fabric could be swapped out for different textures or weights.

The simplicity of the design also means that it can be easily combined with other patterns or designs to create more complex garments. For instance, it could be used as a base for a dress with a skirt pattern attached, or it could be layered under a jacket or cardigan.

In summary, the design you've shown is quite flexible and can be adapted in numerous ways to create a variety of garments, from casual tops to more formal dresses.

Figure 15: Detailed VQA example 1, given an patent image from IMPACT, we design a few questions and use LLaVA to generate answers.

