# OpenReview forum: "IMPACT: A Large-scale Integrated Multimodal Patent Analysis and Creation Dataset for Design Patents"
_NeurIPS.cc/2024/Datasets_and_Benchmarks_Track — NeurIPS 2024 Track Datasets and Benchmarks Poster_

### Official Review · Reviewer_C8Cv · 2024-07-23
**Reviewer C8Cv**

**Rating:** 6
**Confidence:** 3
**Correctness:** This is correct.
**Clarity:** Clarity is sufficient.

**Review:**

While focusing on design patent is interesting, it is unclear how the collected dataset is different from existing crawled datasets. Furthermore, the paper would deserve another round of iteration as many tasks that the dataset promises are only discussed rather benchmarked.

**Strengths:**

* Focusing on design patents is interesting as figures in the patents are more informative.

**Additional Feedback:**

n.a.

**Documentation:**

* Dataset has not been shared, only a sample in the supplementary.
* No mention of where it will be hosted or maintained.
* No license has been documented.
* Code has been shared.

**Ethics:**

The paper mentions that the dataset has "huge potential for novel design inspiration". While this is true, there are huge implications regarding the patent rights for such usage. This topic pertaining to patent inventors has not been discussed at all.

**Limitations:**

This is sufficient.

**Opportunities For Improvement:**

* The paper has unverified claims.
  * The paper claims that the dataset can be used for 3D image reconstruction, visual question answering, design trends, novel design inspiration, etc. None of these have been evaluated and in the current form, it is not possible to conclude whether the dataset has such potential.

* Patents over 14 years are crawled from the US patent office, but only a part of it is used in experiments.
  * Indeed, the classification experiment only uses patents from the last two years, and the retrieval experiment only uses patents from the last 5 years. Why is this the case? Why not use all the crawled patents for experiments? What is the point of collecting all the patents to only use a fraction of it?

* The relevance of captions is unclear.
  * The paper generates captions with a vision-language model. Though, this model has not been made for patents. Furthermore, the captions have not been checked for their relevance. What is the motivation for the captions, and for the method being used to generate them?
  * Furthermore, Table 1 shows that captions are detrimental to performance. Why is this the case? There is no explanation in the paper.

* Differentiation with existing works is unclear.
  * The related work mentions multiple relevant works in the same field. However, it is hard to grasp how the crawled dataset is different wrt each one of them.
  * Is the collected dataset a subset of the Harvard-USPTO dataset [51]? If so, why not create a derivative from it rather than reproducing its methodology and claiming a new name.

* Evaluation is insufficient.
  * In both benchmarks, there are no models dedicated to patents that are evaluated. Only generic Resnet and Vit are evaluated.

**Relation To Prior Work:**

This is sufficient.

**Summary And Contributions:**

The paper focuses on design patents, and proposes to perform patent classification, as well as image/text retrieval.
To achieve this, the paper crawls the US patent office. It extracts relevant patent-related information (e.g., title) and design figures, and generates captions with an vision-language model for each figure. Benchmarks are proposed and show that ViT are better than Resnet for the tasks.

---

> ### Author Rebuttal · Authors · 2024-08-17
>
> **1. The paper has unverified claims. The paper claims that the dataset can be used for 3D image reconstruction, visual question answering, design trends, novel design inspiration, etc. ...**
> **Response:**
>
> Sorry for the confusion. We will clarify these points.
>
> * **3D construction:** We have added 3D construction examples in `the rebuttal PDF` (see Figure 2).
>
> * **VQA examples:** We have already added 2 VQA examples in Appendix A.5 (see Figs. 11 and 12) of the original paper.
>
> * **Design trends/inspiration:** Design Patent data is widely known for design inspiration for industries. It serves as a valuable resource for designers, engineers, and companies for creating products with distinct design. It also helpful so that they don't infringe on existing intellectual property. In this paper, we have organized and elaborated the dataset in way it will be suitable for multimodal learning
>
>
>
> **2. Patents over 14 years are crawled..., but only a part of it is used in experiments...**
>
> **Response:**
>
> We have created the dataset with 14 years of design patents to advance research  as currently research specially on multimodal learning for design patents is limited. We demonstrate the usefulness of the dataset through patent-related tasks even using a subset of the data to show its potential.
>
> Nonetheless, according to the suggestion, we fine-tune the CLIP model on the complete dataset. The results for image-to-text and text-to-image tasks are presented in the table below (I denotes image, T denotes Text, I2T denotes image to text retrival).
>
> |  Model  | I2T  | I2T  | T2I  | T2I    |
> |:-------:|:----:| ---- |:----:| --- |
> |         | R@5  | R@10 | R@5  |  R@10   |
> | RN50 | 23.92  |   32.84  | 25.00 |  34.37   |
> | RN101  | 25.82 |  35.25    | 26.60 |  36.06   |
> | Vit-B  | 28.41  |   38.32   | 29.28  | 39.87    |
> | Vit-B  | 39.59  |   50.44   | 41.72  | 52.55    |
>
>
> **3. The relevance of captions is unclear... Furthermore, Table 1 shows that captions are detrimental to performance...**
>
> **Response:**
> * *Explanation:* To address the lack of detailed textual information in design patents, we aim to generate captions that could enrich the existing data and facilitate multimodal research for design patents. We experiment with three different multimodal models and use LLaVA for caption generation due to its cost-effectiveness, time efficiency, and relevance. To validate the relevance and usefulness of the generated captions, we conduct classification and retrieval tasks which shows that descriptive captions contribute to improved performance.
>
> * *Table 1:* We humbly disagree with the reviewer. Table 1 shows that the detailed captions are useful for patent classification, where the combination of images, titles, and captions yields the best results. This shows that captions are helping in patent classification.
>
> * *Additional elements:* We have added *human evaluation* for the generated captions (response 1 for Reviewer pWTK). We have also added examples of the generated captions from three different models in the `rebuttal pdf`.
>
>
> **4. Differentiation with existing works is unclear. ... Is the collected dataset a subset of the Harvard-USPTO dataset [51]? If so, why not create a derivative from it ...**
>
> **Response:**
>
> Existing design patent datasets and how they differ from our dataset are described below:
>
> 1. Deeppatent [1] is limited in the number of patents. It included data from 2018-2019 only. Does not include any captions or descirptive information.
> 2. Deeppatent2 [2] contains data from 2007 to 2020 (In our dataset, we include 2007 to 2022). This dataset does not include informative caption. Thus, the opportunity of performing multimodal tasks is limited as design patents usually do not have informative text. We address this issue with descriptive captions that includes the shape, functionality of a particular design patent to facilitate multimodal learning. Thus, we are able to employ multimodal models on our dataset as can be seen from the experimental results.
>
> All the other datasets including Harvard-USPTO dataset [3] is Utility patent dataset and does not include any image at all. Furthermore, utility patents are different from design patents and usually have a large amount of texts including abstracts, claims, descriptions, etc. **Our dataset is not a derivative of Harvard-USPTO dataset. IMPACT is a dataset with Design patents from 2007-2022.**
>
> References:
> [1] Kucer et al. DeepPatent: Large scale patent drawing recognition and retrieval. CVPR, 2022.
> [2] Ajayi, K. et al. DeepPatent2: A Large-Scale Benchmarking Corpus for Technical Drawing Understanding. Scientific Data, 2023
> [3] Suzgun et al (2024). The harvard uspto patent dataset: A large-scale, well-structured, and multi-purpose corpus of patent applications. NeurIPS.
>
>
> **5. Evaluation is insufficient. In both benchmarks, there are no models dedicated to patents that are evaluated. Only generic Resnet and Vit are evaluated.**
>
> **Response:**
>
> Multimodal models are predominantly trained on natural images, hence, they often struggle to accurately represent the intricacies of design images, which are typically black-and-white sketches. **Currently, there are no multimodal models dedicated to design patents.** To address this, we implement CLIP and use Vit-B-32 as backbone. We further fine-tune these backbones to better adapt them to our design patent dataset.
>
> Image retrieval results (mAP) comparison between our fine-tuned PatentCLIP and other SOTA patent models are given below. The table shows that CLIP, when fine-tuned on the Impact dataset, outperforms other models and achieves the highest mAP.
>
>
> |        Model        |  mAP  |
> |:-------------------:|:-----:|
> | ViT-B + ArcFace [1] | 0.614 |
> | CLIP-ViT-B + ArcFace| 0.645 |
> | **PatentCLIP-ViT-B + ArcFace** |  **0.657** |
>
>
> [1]Higuchi, K., & Yanai, K. (2023). Patent image retrieval using transformer-based deep metric learning. World Patent Information, 74, 102217.

---

> > ### Author Response · Authors · 2024-08-23
> > **Waiting for your feedback**
> >
> > Dear Reviewer,
> >
> > We sincerely appreciate your thorough review and valuable feedback on our paper. We hope our rebuttal has addressed your concerns. If you have any further questions, please let us know. We look forward to your feedback.
> >
> > Regards,
> >
> > Authors

---

> > > ### Author Response · Authors · 2024-08-27
> > > **A gentle reminder**
> > >
> > > Dear Reviewer C8Cv,
> > >
> > > Thanks for your constructive reviews and effort. We are approaching the end of the discussion period, and thus, we would greatly appreciate your feedback. We hope our rebuttal has addressed your concerns. Please let us know if there are any other questions. We are eagerly waiting to hear from you.
> > >
> > > Regards,
> > >
> > > Authors

---

> > > > ### Comment · Reviewer_C8Cv · 2024-08-28
> > > > **Response**
> > > >
> > > > Thank you for the comprehensive rebuttal. It addresses most of the points I have raised and I am happy to upgrade my score.
> > > >
> > > > A couple of points:
> > > > * Captions. The point was more about the setting where caption is not used in combination with other modalities.
> > > > * Differentiation with existing work. It would be beneficial for the paper to mention how the crawling methodology related to existing work. It is fine with me if the methodology is the same but only differs by the type of patents.

---

> > ### Author Response · Authors · 2024-08-29
> > **Thanks!**
> >
> > Thank you for your support, and we truly appreciate it. We note your points. Thank you for the feedback. We will incorporate these suggestions in the revised version.
> >
> > We add these notes to the discussion for completeness.
> >
> > **1. Captions:** Thanks! We note your point.
> >
> > **2. Differentiation with existing work of crawling methods:** Thanks for this point.  We download raw patent data from the USPTO Bulk Data Storage System (the same **source** used by existing works). However, we have preprocessed and cleaned the raw data **differently**. For example, the XML files of design patents are different from utility patents (e.g., Harvard-USPTO dataset). Harvard-USPTO has also considered inventor-submitted versions of the patents, where as we have considered the granted patent documents. We have parsed all the plain text and extracted the fields from the XML files of the design patents and stored in the output csv files. DeepPatent and DeepPatent2 have also crawled the data from the same source. However, we have generated the captions with patent titles and images which are currently missing in the previous datasets.

---

### Official Review · Reviewer_H2SD · 2024-07-23

**Rating:** 7
**Confidence:** 3
**Correctness:** Yes.
**Clarity:** Yes.

**Review:**

### Strengths
1. The paper is well-organized and easy to follow.
2. The discussions of background and related work on patent datasets are comprehensive.
3. The proposed dataset fills a gap in existing patent data, which often lacks detailed descriptions necessary for multimodal tasks like classification and retrieval.
4. All the data and code are available via an anonymous link.

### Weaknesses
Overall, I am satisfied with this paper, except for certain experimental details that may affect the perceived difficulty of the task.
1. The paper uses LLaVA for caption generation and claims that LLaVA performs better than GPT-4 (L205), which contradicts intuition. There are more advanced multimodal models available, such as Qwen-VL, GPT-4v/o, etc. It is suggested to use different multimodal models for caption generation and comparative discussion to see the impact of caption quality on retrieval and classification.
2. The backbone models used in the classification and retrieval tasks are somewhat limited. Consider incorporating more advanced models to enhance the quality of the paper.

**Strengths:**

See **Strengths** in #Review.

**Additional Feedback:**

n/a

**Documentation:**

Yes.

**Ethics:**

No.

**Limitations:**

Yes.

**Opportunities For Improvement:**

See **Weaknesses** in #Review.

**Relation To Prior Work:**

Yes.

**Summary And Contributions:**

This paper presents IMPACT, a large-scale multimodal patent dataset with detailed captions for design patent figures, including half a million design patents with 3.6 million figures and captions from the US Patent and Trademark Office spanning 2007 to 2022. Preliminary evaluations show that integrating images with generated captions significantly enhances model performance on multimodal tasks like classification and retrieval. The paper also proposes two underexplored tasks for future research using IMPACT as a benchmark: 3D Image Construction and Visual Question Answering.

---

> ### Author Rebuttal · Authors · 2024-08-17
>
> **1. The paper uses LLaVA for caption generation and claims that LLaVA performs better than GPT-4 (L205), which contradicts intuition. There are more advanced multimodal models available, such as Qwen-VL, GPT-4v/o, etc. It is suggested to use different multimodal models for caption generation and comparative discussion to see the impact of caption quality on retrieval and classification.**
>
>
>
>
> **Response:** We apologize for the confusion. LLaVA uses CLIP as the backbone for their visual model and GPT-4 as part of their methodology. GPT-4 was employed to generate instruction following data which are then used to fine-tune other models. They provided an example showing that LLaVA offers a more comprehensive response than GPT-4.
>
> We incorporate  both  Qwen-VL and GPT-4v/o to generate captions for 1000 samples from the data of 2022. We consider three factors for chosing caption models, including (1) runtime (2) costs and (3) quality of the captions.
>
> **(1) Runtime:** GPT4-o, LLaVA, and Qwen-VL required an average of 4.7, 3.98, and 3.25 seconds to generate each caption, respectively.
>
> **(2) Costs:** In terms of cost, generating 174 captions with GPT incurred a charge of $1, while the other two models, LLaVA and Qwen-VL  were free of charge.
>
> **(3) Quality of the captions:**
> * Qualitative  analysis examples are shown in rebutal PDF Figure 1 (a). We observe that Qwen is not able to generate function descriptions in many cases. It also inserts Chinese words into  English captions in some cases.
> * We conduct zero-shot multimodal retrieval tasks to evaluate the quality of captions (please see table below). GPT-4o performs the best and LLaVA has better performance than Qwen. Considering all the above factors, we use LLaVA as our captioning model.
>
> *Therefore, considering all these factors, we use LLaVA as caption model to generate captions for IMPACT dataset which contains more than 435,000 patents.*
>
>
> |  Model  | I2T  | I2T  | T2I  | T2I    |
> |:-------:|:----:| ---- |:----:| --- |
> |         | R@5  | R@10 | R@5  |  R@10   |
> | Qwen-VL | 6.6  |   10.8   | 9.4  |  13.3   |
> | GPT-4o  | 10.5 |  15.3    | 12.4 |  17.5   |
> |  LLaVA  | 8.4  |   12.2   | 9.5  | 13.8    |
>
>
>
>
>
> **2. The backbone models used in the classification and retrieval tasks are somewhat limited. Consider incorporating more advanced models to enhance the quality of the paper.**
>
> **Response:**
>
> Thanks for the suggestion!
>
> * **Classification:** As per your suggestions, we use CLIP (the popular multimodal model) with a Vit-B-32 backbone to classify patents in two configurations: image-caption and image-title using fine-tuned CLIP. The table shows the accuracy produced by the CLIP model in patent classification.
>
> |  Metric  | Model         |
> |:-------:|:--------------:|
> |         |    CLIP        |
> | Acc.(Image+Caption)    |    22.72            |
> | Acc.(Image+Title)    |          22.42      |
>
>
> * **Retrival:** For retrieval, we believe CLIP as the standard model for multimodal task and we have already reported the results in Table 2 in the original paper.

---

> > ### Author Response · Authors · 2024-08-23
> > **Thanks!**
> >
> > Dear Reviewer,
> >
> > We sincerely appreciate your positive review and support for our work. If you have any further questions, please let us know.
> >
> > Regards,
> >
> > Authors

---

### Official Review · Reviewer_pWTK · 2024-07-25

**Rating:** 7
**Confidence:** 4
**Correctness:** The paper is technically sound
**Clarity:** The paper is very well written

**Review:**

Overall this is a good paper and datasets, which can be useful to the community.

**Strengths:**

The paper is clearly written, and the dataset is of good quality overall. Experiments are technically sound and show two uses of the dataset on tasks of interest. The images are different from standard computer vision datasets, which makes the domain gap an interesting feature of the dataset.

**Additional Feedback:**

N/A

**Documentation:**

Details are provided

**Ethics:**

No ethical problems detected

**Limitations:**

See above

**Opportunities For Improvement:**

I would have liked to see some more discussion on the quality of the generated captions from LLaVA, ideally providing some human evaluation of the accuracy of such descriptions.

Another point is that the discussion of the two additional tasks (3D reconstruction and VQA) looks limited. It would have been better to add some examples.

**Relation To Prior Work:**

Previous work is extensively discussed.

**Summary And Contributions:**

The paper presents a dataset of design patents from 2007 to 2022 in the US. Each patent has one or more images together with a title and other metadata. Additionally, synthetic descriptions are generated with LLaVA and included in the dataset. The use of the dataset is demonstrated with experiments on two tasks: classification and multimodal retrieval. Additionally, the paper discusses the use of the dataset in two other tasks: 3D reconstruction and VQA.

---

> ### Author Rebuttal · Authors · 2024-08-17
>
> **1. I would have liked to see some more discussion on the quality of the generated captions from LLaVA, ideally providing some human evaluation of the accuracy of such descriptions.**
>
> **Response:** Thanks! As per suggestion, we have conducted a preliminary human evaluation to assess the generated captions from LLaVA. In the study, we have a total of 12 participants. All participants are graduate students in STEM with prior research experience. Each participant reviews three sets of titles, captions, and images. We ask the user to read these carefully and determine if the captions can add value to the image as a descriptive caption in their opinion with the following questions: *Did the caption describe the image correctly? Did the caption can describe the shape and functionality of the image logically?* As a result, in more than 60% of their responses, they agree with the quality of the generated captions. However, one of the examples shows the limitations of general multimodal captioning models and we would consider this as a future direction.
>
>
> We include the examples in Figure in `the rebuttal PDF` (see Figure 1b).
>
>
> Moreover, we provide experimental results on generated captions by three models (see Reviewer H2SD Response 1).
>
>
>
> **2. Another point is that the discussion of the two additional tasks (3D reconstruction and VQA) looks limited. It would have been better to add some examples.**
>
> **Response:**
>
>
> * For VQA, we provide 2 examples in our Appendix (see Figure 11 and 12). We have shown two examples with 6 questions and answers. We design a set of questions for IMPACT dataset, such as "any new design suggestions for patent_title". We integrate LLaVA to generate answers and we observe the VQA system can provide helpful information for further patent analysis. Thus, we propose IMPACT-VQA as a future work.
>
> * For 3D reconstruction, we provide 2 examples in `the rebuttal PDF` (see Figure 2). We utilize ControlNet [1] to generate 3D photos for patent sketches. Comparing the results of prompting with IMPACT captions and prompting with patent title, we observe that our captions can provide more guidance for diffusion models.
>
> [1] Zhang, Lvmin, Anyi Rao, and Maneesh Agrawala. "Adding conditional control to text-to-image diffusion models." Proceedings of the IEEE/CVF International Conference on Computer Vision. 2023.

---

> > ### Author Response · Authors · 2024-08-23
> > **Thanks!**
> >
> > Dear Reviewer,
> >
> > We sincerely appreciate your positive review and support for our work. If you have any further questions, please let us know. Please see the rebuttal pdf for the suggested examples.
> >
> > Regards,
> >
> > Authors

---

> > > ### Comment · Reviewer_pWTK · 2024-08-30
> > >
> > > Thank you for the response and clarifications. I am happy to keep my score and support the paper for acceptance.

---

### Author Rebuttal · Authors · 2024-08-17

We would like to thank the reviewers for their constructive feedback. We have provided a comprehensive point-by-point response to their comments. The major points include:

1. Empirical Study:
   - Results on three models for caption generation (Reviewer H2SD)
   - The results from CLIP for patent classification (Reviewer H2SD)
   - Results from the fine-tuned CLIP model on the complete dataset (Reviewer C8Cv)
   - Image retrieval results (mAP) comparison between our fine-tuned PatentCLIP and other SOTA patent models (Reviewer C8Cv)


2. Added discussion:
   - Human evaluation (Reviwer pWTK)
   - Examples of 3D construction (Reviwer pWTK)

---

### Decision · Program_Chairs · 2024-09-26

**Decision:**

Accept (Poster)

**Comment:**

This paper presents a high-quality dataset that addresses a gap in existing patent data for multimodal tasks like classification and retrieval. The experiments are technically sound, showcasing two applications of the dataset, and the unique images provide an interesting domain gap. The background discussions are comprehensive, and all data and code are available via an anonymous link. However, the claim that LLaVA outperforms GPT-4 in caption generation is counterintuitive, especially given the availability of more advanced multimodal models like Qwen-VL and GPT-4v/o. The authors should explore these alternatives for caption generation and consider incorporating more advanced backbone models to enhance the study. Overall, while strong, the paper could benefit from addressing these points to improve its impact.